# The Development and Validation of a Satisfaction and Emotional Perception Scale for Women Undergoing Fertility Treatment

**DOI:** 10.3390/healthcare13192416

**Published:** 2025-09-24

**Authors:** Laura de la Torre García, Andrés Arias Astray, Iria Osa Subtil, Concepción del Pino Ortega, Reyes Velázquez Barbado, Carlos González Duque, María José Dios-Duarte

**Affiliations:** 1Health Service of Castilla y León (SACYL), Río Hortega University Hospital, 47007 Valladolid, Spain; 2Social Work Department, Faculty of Social Work, Institute of Technology of Knowledge, Complutense University of Madrid, 28223 Madrid, Spain; aariasas@ucm.es; 3Faculty of Medicine, Health and Sport, Department of Medicine, European University of Madrid, 28108 Madrid, Spain; iria.delaosa@universidadeuropea.es; 4Health Service of Castilla y León (SACYL), University Clinical Hospital of Valladolid, 47007 Valladolid, Spain; 5Minsait—An Indra Company, 47006 Valladolid, Spain; 6Nursing Department, Faculty of Nursing, University of Valladolid, 47005 Valladolid, Spain

**Keywords:** patient satisfaction, nursing care, Emotional Self-Perception, fertility

## Abstract

**Background/Objectives**: It is known that assisted human reproduction in infertility causes psychological and mental instability in those undergoing this treatment. Patient satisfaction, personal experiences with assisted reproduction treatment (IVF) and self-perception of emotional distress require a comprehensive study using complete and scientifically validated instruments. The aim of this study was to develop and validate a scale enabling nurses to assess patient satisfaction with the care they receive, personal satisfaction with the assisted reproduction treatment process, and self-perceptions of emotional discomfort. **Methods**: This study employed a mixed research strategy. Initially, a systematic literature review informed the qualitative phase, which involved expert focus groups in formulating the questionnaire items. Subsequently, the developed scale underwent psychometric analysis in the quantitative phase and was given to women undergoing fertility treatment. **Results**: The scale was found to have an acceptable level of factorial validity and reliability. The items were consistent and homogeneous, with high saturation in their respective factors (3). A negative covariance was observed between factors 1 and 2, and between factors 2 and 3, along with a positive covariance between factors 1 and 3. Furthermore, it was found that satisfaction with nursing care (F1) was associated with a lower need for professional psychological help and that a greater perception of emotional well-being (F2) was associated with a greater need for this type of help (F1: β = −0.07, *p* = 0.002; F2: β = 0.10, *p* = 0.004). **Conclusions**: This scale is a robust and dependable instrument, demonstrating its validity and reliability. Most notable are its user-friendly nature, ease of administration, and minimal time needed. Moreover, the scale proves effective in identifying women who require professional psychological support, which is a critical distinction with significant implications for patient care. In practical terms, the scale equips nurses with a powerful tool for conducting a thorough and efficient assessment of women undergoing fertility treatment.

## 1. Introduction

### 1.1. What Is Infertility?

Infertility is defined as the inability to achieve a clinical pregnancy after 12 or more months of regular unprotected intercourse [1].

According to the World Health Organization, 17.5% of the world’s adult population is infertile. Infertility rates are similar across all countries, regardless of their level of wealth and development: 17.8% in high-income countries and 16.5% in low- and middle-income countries [2]. Infertility affects between 17 and 26% of childbearing-age couples (3% of the world’s population), or about 48.5 million couples. Therefore, infertility is a global health problem that is only increasing in both developing and developed countries [3].

Infertility is caused by inadequate functioning of the reproductive tract and, in some cases, by chronic diseases and sexual conditions that are incompatible with intercourse [4].

In recent decades, advances in assisted reproductive technology (ART) have increased treatment options, but they have also created new emotional, ethical, and social challenges [1].

### 1.2. Treatment Options

Assisted reproductive technologies (ARTs) include all interventions involving the in vitro manipulation of oocytes, sperm or embryos for reproductive purposes, such as in vitro fertilization (IVF), intracytoplasmic sperm injection (ICSI), gamete intrafallopian transfer (GIFT), pre-implantation genetic testing (PGT), cryopreservation of gametes and embryos, gamete and embryo donation, and surrogacy [5].

Although these options have significantly increased the chances of achieving pregnancy, they do not completely mitigate the emotional impact of this process.

The prevention and treatment of infertility is part of comprehensive sexual and reproductive health services available for a full and satisfying life. At the same time, it must be remembered that infertility treatment involves not just biological or physical factors but also psychological and social factors [6].

Infertility treatment is usually long-term and involves several consultations. Importantly, this treatment is not always successful. All of these factors affect the emotional state of the person undergoing treatment, often impacting their mental and physical health and, consequently, the fertility treatment itself [7,8,9].

### 1.3. Psychological, Emotional and Social Difficulties Related to Infertility

The psychological impact of infertility can reach levels similar to those experienced in serious chronic diseases such as cancer, heart disease or HIV infection. In fact, many patients undergoing fertility treatments report that this process is one of the most painful and stressful experiences of their lives [10].

The perception of self-image in infertile individuals tends to deteriorate, manifesting itself through emotional instability (unregulated emotions) and the onset of severe depression [11].

Very few studies have investigated the psychosocial and emotional impact on people with infertility problems. However, we would highlight some studies shown that the diagnosis of infertility itself can cause emotional distress, anxiety, and depression, with these effects being more prevalent and severe in women [7,8,9].

A study conducted in 2011 [12] revealed that women undergoing medical treatment for infertility showed higher levels of depression, employed more avoidant and emotional coping styles, and scored lower on scales of acceptance and self-compassion compared to a control group without fertility issues. The study by Martins et al. (2014) also showed that social support in the case of women with infertility contributes to a lower perception of stress and vice versa [13].

In general terms, research conducted on individuals and couples facing infertility shows that women are more vulnerable to developing negative consequences for their mental health than men [14,15].

Thus, if they seek fertility treatment, these psycho-emotional disturbances will be exacerbated by those that arise as a result of their treatment.

### 1.4. Psychoemotional and Social Problems Related to Infertility Treatment

The psycho-emotional impact of infertility may differ from that directly linked to its treatment. However, it is very difficult to distinguish between the two. Research on this subject is scarce. Nevertheless, we highlight the study conducted by Galhardo, Cunha and Pinto-Gouveia (2011), which showed that women undergoing medical treatment for infertility exhibited higher levels of depression, employed more evasive and emotional coping styles, and scored lower on scales of acceptance and self-compassion compared to a control group without fertility problems [12]. Also, the study conducted by Renzi et al. [16] demonstrated that anxiety plays a crucial role in diminishing the perception of emotional, mind–body, social, environmental, and tolerable quality of life. Elevated anxiety levels were found to have adverse effects on emotions, physical health, cognition, behaviour, social aspects, and treatment outcomes. For instance, women exhibited more negative perceptions regarding the accessibility and quality of health services. The same research affirmed that perceptions of affection and care had a direct mitigating impact on the effects of infertility.

In light of this overview, jointly assessing and monitoring the psycho-emotional area is a fundamental step in treating these individuals.

### 1.5. Assessment of the Psycho-Emotional State and Improved Treatment Outcomes

Research has demonstrated that negative emotional states can extend the duration of infections and impede wound healing [17,18], contribute to disease relapse [19], or heighten susceptibility to various illnesses [20]. Regarding emotional states, numerous studies have established their influence on the frequency of somatic symptoms, the onset of physical and mental health issues, the increased utilization of healthcare services, and the adoption of unhealthy lifestyles [21]. Furthermore, emotional states are correlated with patient satisfaction in treatment outcomes [22] and overall life satisfaction [23].

A study conducted among women undergoing infertility treatment revealed that a formal communication strategy devoid of emotional aspects induces stress in those undergoing treatment [24].

Therefore, it is understood that to achieve better results in infertility treatments, it is advisable to consider these aspects. This will make patients undergoing treatment feel supported and will also generate beneficial effects from possible interventions on psycho-emotional variables.

### 1.6. Assessment of Psycho-Emotional Difficulties and Level of Satisfaction: Areas for Improvement

Infertility, along with the treatments used to address it, is a significant source of stress, characterized by its high intensity, its persistence over time, and patients’ perception of limited control. This combination of factors makes coping particularly complex, affecting both the psychological well-being and quality of life of those going through these processes.

Satisfaction is a variable that directly influences both quality of life and adherence to treatment [25,26].

Satisfied patients are more likely than dissatisfied patients to use healthcare services consistently, maintain relationships with specific healthcare providers, and adhere to prescribed treatment regimens. This increases the chances of success [27,28] and therefore results in patients not abandoning treatment.

Patient satisfaction, in turn, serves as a metric for assessing the quality of care delivered by health professionals to the extent that it has been characterized as a measure of effectiveness and even considered an objective control when studying the quality of health services [29]. It can therefore be deduced that satisfaction is related to the patient’s continuity of treatment and to a better state of well-being, contributing to a better adjustment to the situation.

Infertility tends to cause those affected to feel trapped and vulnerable, characterized by emotions of threat, frustration and failure in achieving an important life goal. This perception also has implications for self-esteem, relationship dynamics and social integration, increasing the psycho-emotional impact of the process [11].

Patient-centred care takes a holistic view of the individual, considering their values, expectations, and emotional needs. In the field of infertility treatment, this approach is vital for alleviating the psychological impact of treatment and enhancing the patient experience.

### 1.7. Role of Nursing in the Treatment of People with Infertility Problems

Nurses play an essential role in addressing infertility by providing emotional support, education, and ongoing accompaniment throughout the therapeutic process. Their intervention facilitates understanding of procedures, promotes self-care, and allows for early detection of signs of anxiety or depression. In addition, nurses are a key link within the interdisciplinary team, facilitating communication between patients, doctors and psychologists, and ensuring that care is truly person-centred. This role of proximity and emotional support makes nursing a strategic pillar in ensuring quality care in infertility.

Nursing interventions for patients undergoing assisted human reproduction treatment not only contribute to improving patients’ health but also have a significant impact on the care they receive. Notably, in most instances, professionals in direct contact with patients identify their needs through the information shared during consultations at Assisted Human Reproduction Units (URHA).

Nurses specializing in assisted reproduction highlight that these patients often express emotional instability during consultations, leading to complaints that may be interpreted as a lack of understanding and inadequate care. Therefore, there is a pressing need for a measurement tool that enables the assessment of both patient satisfaction and emotional states during assisted human reproduction treatments. The final aim is to enhance the care provided and elevate the overall quality of healthcare.

Based on the above, we consider it essential to measure patients’ satisfaction with the care they receive, the progress of their treatment, and their emotional well-being.

The objective of this study was to develop and validate a scale that enables nurses to assess patients’ satisfaction with the care they receive, their personal satisfaction with the assisted reproduction treatment process, and their self-perception of emotional discomfort. This implies providing comprehensive, patient-centred care.

This work focuses on women because research has shown that they suffer more psychosocial and emotional problems related to infertility.

## 2. Materials and Methods

### 2.1. Design

This study employed a mixed research strategy. Initially, a systematic literature review informed the qualitative phase, which involved expert focus groups in formulating questionnaire items. Subsequently, the developed scale underwent psychometric analysis in the quantitative phase and was given to women undergoing fertility treatment.

### 2.2. Participants

The expert group was purposefully sampled to consist of eight actively engaged nurses working in assisted reproduction clinics. All members held a university degree in nursing with over five years of practical experience. Each participant willingly and impartially participated in the study.

The patient group for scale validation comprised 97 women selected through convenience sampling from assisted reproduction clinics. Participants aged 18 to 40 voluntarily agreed to participate, initiate treatment, and attend three or more clinics. Proficiency in Spanish, comprehension of the study’s aims, and no history of fertility preservation treatment were additional criteria.

The exclusion criteria were the presence of any chronic organic disease, a psychological disorder diagnosed by a psychologist, and/or a psychiatric disorder diagnosed by a psychiatrist.

### 2.3. The Procedure and Development of the Scale

To search the relevant literature, we utilized search engines, including PubMed, Dialnet, Cochrane, and ScienceDirect. Keywords such as “assisted human reproduction”, “psychosocial factors”, “social factors”, and “psychological factors” were employed with the Boolean operator “AND” to ensure specificity.

Focus groups were formed with experienced nurses to gather information, and based on the results, a comprehensive set of premises for the questionnaire was developed.

Throughout October and November 2021, a total of four biweekly meetings were held to define the items of the scale, with expert nurses contributing valuable insights to shape its content. The discussions were guided by open questions, focusing on patients’ needs and their perceptions of care. These sessions, lasting a maximum of 2.5 h with a break, specifically addressed humanistic and psychosocial aspects.

The following open-ended questions were prepared to initiate discussions among the expert nurses:

1. From a humanistic standpoint, with a focus on the psychosocial aspect, what needs do you observe in patients undergoing assisted human reproduction treatment?

2. Based on your experience, how do you envision enhancing the care provided to these women from a holistic perspective?

3. In your opinion, how do patients define and perceive the care they receive from their caregivers?

In the initial three meetings, professionals analyzed patients’ comments and observed signs and symptoms during consultations to identify three factors crucial for women undergoing assisted reproductive treatment. The following factors were derived from nurses’ comments based on clinical experience: anxiety, evidenced by expressions like “I am very worried,” and associated symptoms such as a dry mouth, tachycardia, and increased blood pressure. The second factor, mood affectation, often accompanies anxiety, with patients expressing feelings like “I can’t take it anymore” and exhibiting signs of depression such as insomnia and lack of concentration. The third factor, dissatisfaction, posed challenges in identification due to varying patient complaints. In a subsequent meeting, the nurses collaboratively refined a 20-item scale, eliminating items that were difficult to understand, not focused on what was to be measured, or duplicated, to produce the final 11-item scale detailed in Appendix A. A scale was developed with affirmative Likert-type statements ranging from 0 (‘strongly disagree’) to 10 (‘strongly agree’). The items eliminated were as follows:

“I think that the emotional support provided by the nurse was adequate and effective because it considerably reduced my distress”. This item was eliminated because the expert group considered it to be convoluted and not focused on anxiety. It was decided to keep item 10. “I feel that the emotional support provided by the nurse was sufficient to reduce my level of anxiety”.

“I feel that the nurse responsible for my treatment demonstrates a deep understanding of my situation and the difficulties I am going through.” Item 7 expresses the same idea, and we considered it to be simpler; therefore, this item was removed, and item 7 was retained. “I feel that the nurse in charge of my treatment understands what I am going through”.

“I feel supported and understood by my loved ones in the face of the difficulties I am experiencing in the reproductive sphere”. This item was removed, and item 2 was retained because it was considered a duplicate. “I am satisfied with the support I receive from my family and/or friends about my reproductive problems”.

“The care provided by the nurse in charge of the Assisted Human Reproduction Unit (AHRU) satisfactorily meets what I consider to be relevant to my care”. This item was considered to be duplicated with item 6 and was therefore deleted. Item 6. “I have the care of the Assisted Human Reproduction Unit (AHRU) nurse I want”.

“My level of satisfaction is positive with regard to the quality of the services offered, insofar as they have adequately responded to my emotional needs”. This item was removed because item 8 expresses the idea more simply. “I am satisfied with the quality of the services available to me to meet my emotional needs”.

“The complications related to my reproductive health have had a negative impact and caused problems in the dynamics and quality of my romantic relationship”. This item was removed and replaced with item 4. “My reproductive problems have had a negative impact on my relationship with my partner”. The explanation for this is that item 4 is easier to understand.

“The reproductive treatment process has a negative effect on my emotional stability and psychological well-being”. Instead, item 5 was considered because it was more focused on mood, which was what we wanted to measure specifically. “Reproductive treatment has a negative effect on my mood”.

“The guidance provided by the nurse throughout the process has contributed to a significant reduction in my level of anxiety regarding reproductive treatments”. This item was removed as it was considered to duplicate item number 11. “After receiving the information provided by the nurse during the process, my level of anxiety about the reproductive treatments has decreased”.

“I consider that the information received regarding the medicines administered, both current and previous, as well as on the treatments in progress or already completed, has been relevant and sufficiently complete”. This item was considered difficult to understand due to the way it was expressed, so it was decided to keep item 9. “The information I received about the medicines I am taking or have taken and/or treatments I am receiving or have received was adequate”.

By removing these items, we wanted to make the scale clearer, simpler, and easier to answer honestly. In addition, by making the scale shorter, we wanted to maintain the attention of the women who completed it.

To assess the validity and reliability of the scale, it was given to women receiving assisted reproduction treatment. Participants, recruited through clinic appointments, provided voluntary and disinterested consent.

The subsequent fertility treatment involved in vitro fertilization (IVF). Patients were initially briefed on the procedural steps, encompassing ovarian stimulation, oocyte retrieval, insemination and fertilization, embryo culture, and embryo transfer. The patient’s explicit consent to the proposed treatment was documented. Continuous support was offered through scheduled consultations to address any uncertainties the patient might encounter. In the event of successful treatment, the initial pregnancy consultation was conducted, and the patient was subsequently referred to a gynecologist. Conversely, if the treatment proved unsuccessful, psycho-emotional support was extended.

### 2.4. Ethics

The study was conducted from September 2021 to July 2022 and approved by the Research Ethics Committee of the Valladolid East Health Area of the Clinical Hospital of Valladolid (Ref. PI 22-2546). Reporting was carried out using the STROBE and SRQR checklists.

### 2.5. Statistical Analysis of the Scale

First, descriptive statistics were calculated (the mean, standard deviation, skewness, and kurtosis measures). In addition, the discrimination index based on correlations was calculated to identify items that did not contribute to the scale’s evaluation (<0.30 is not explanatory) [30]. Secondly, to test dimensionality, confirmatory factor analysis was carried out on three correlated factors using the method of Diagonally Weighted Least Squares (DWLSs). To assess the fit of the proposed models, the following indices were calculated: a chi-square ratio (χ^2^/df) ≤ 3, a comparative fit index (CFI) and Tucker–Lewis index (TLI) ≥ 0.95 and 0.90, a root mean square error of approximation (RMSEA) and standardized root mean square (SRMR) ≤ 0.08 and 0.10, respectively [31,32,33]. Thirdly, the reliability of the scale was assessed, and adequate values were equal to or greater than 0.70 [34]. In addition, the predictive validity of the scale was tested in relation to the level of anxiety and the need for professional support using multiple linear and logistic regression models. Their quality was evaluated using the corrected coefficient of determination and Nagelkerke’s coefficient, respectively. Finally, to determine the diagnostic performance and the need for professional psychological support, AUC values (area under the curve) were calculated using the ROC test, with acceptable classification values of > 0.70 [35]. The analyses were carried out using RStudio (4.2.3).

## 3. Results

### 3.1. Descriptive Analysis of the Items and Reliability

The items’ means and standard deviations were between 2.73 and 8.23 and 1.97 and 3.32. The shape of the distributions indicates that some variables deviated from normality, especially Items 2, 3, 4, and 9, which showed extreme skewness and extreme kurtosis. Items 2, 3, and 9 showed negative skewness and leptokurtic skewness, meaning that most subjects scored high on these variables. Item 4 presented a positive asymmetric and platykurtic distribution, meaning that most participants scored low on this variable. The remaining variables had a more normal shape. The Discrimination Index showed high relationships for the first factor, moderate correlations for the second factor, and low associations with the third factor (Table 1).

Internal consistency analyses showed high reliability (ω = 0.91) for Factor 1 and adequate reliability (ω = 0.78) for Factor 2. Factor 3 did not achieve sufficient values (ω = 0.55).

### 3.2. Confirmatory Factor Analysis

Metrics were fixed for the most saturated items (Items 7, 5, and 2). The fit results showed controversial data (χ^2^/df = 3.2; *p* < 0.001; CFI = 0.99; TLI = 0.98; RMSEA = 0.15 with CI 90% [0.12, 0.18]; SRMR = 0.08). On the one hand, the fit was excellent for CFI and TLI and adequate for SRMR and the chi-square ratio. On the other hand, the RMSEA did not reach an adequate value. To interpret the values of the fit indices, it is important to consider the sample size, the complexity of the model, and the saturation of the items. Considering the high saturations of the items in their corresponding factors and medium sample size, a low RMSEA can be used [35]. Therefore, the model was accepted as valid (Figure 1).

The model’s latent variable analysis showed a strong relationship between the variables observed and their corresponding factors, especially for Factor 1. In addition, a negative covariance was observed between Factors 1 and 2, and between Factors 2 and 3, alongside a positive covariance between Factors 1 and 3.

### 3.3. Predictive Validity

In terms of predictive validity, the model explains 53% of the variance in anxiety levels, making it a good predictor of this variable (F = 36.0, *p* < 0.01). Perceived emotional discomfort had a statistically significant positive effect on perceived anxiety (β = 0.27, *p* < 0.01). On average, for every one-point increase in perceived emotional discomfort, anxiety increased by 27 points (Table 2).

On the other hand, the logistic regression model correctly classified 78.0% of the cases that did not need professional psychological help (39 out of 50) and 69.8% of the cases that needed professional psychological help (30 out of 43) using a cut-off point of 0.5. The model explains approximately 41% of the variance in the dependent variable (R^2^N = 0.41) and is a good predictor of the need for professional psychological help (χ^2^ = 34.0, *p* < 0.01).

Greater satisfaction with nursing care (F1) was associated with a lower reported need for professional psychological help, while a greater perception of emotional distress (F2) was associated with a greater need for such help (F1: β = −0.07, *p* = 0.002; F2: β = 0.10, *p* = 0.004).

### 3.4. ROC Curve

Finally, the performance of the scale in diagnosing the need for professional psychological help was determined using an ROC curve (see Figure 2). The model has an accuracy of 74.2%, a specificity of 78.0%, and a sensitivity of 69.8%, with a cut-off point of 0.5.

## 4. Discussion

### 4.1. Patient-Centred Care and the Role of the Scale

The study aimed to address a gap in the literature by developing and validating the Satisfaction and Emotional Perception in Assisted Reproduction Scale (SEPAR). The quality of patient care, personal satisfaction with assisted reproduction treatment (IVF), and self-perception of emotional discomfort required extensive study using comprehensive and scientifically validated instruments [36,37]. As healthcare places increasing importance on patient-centred approaches, it is essential to have tools that enable us to understand patients’ perspectives. Based on this, we can then contribute to improving the quality of care and the biopsychosocial well-being of those receiving treatment. In this context, it is well documented that infertility and its treatments generate high levels of stress and anxiety in patients, which reinforces the need to incorporate instruments that facilitate the identification and resolution of these specific issues [10].

### 4.2. Scale Utility and Structure

The SEPAR scale offers a practical solution for nurses to efficiently assess patient satisfaction, personal experiences with IVF, and emotional discomfort within the constraints of a clinical setting.

The three-factor structure of the scale showed evidence of validity and internal consistency in two of them, explaining a significant proportion of the observed variance. Although the incremental fit indices (CFI and TLI) were excellent and the SRMR was within acceptable values, the RMSEA remained above the recommended thresholds, which suggests that the results should be interpreted with caution and the model verified in future replications. The RMSEA is an index that is particularly sensitive to sample size and model complexity and tends to inflate in small samples with few degrees of freedom, falsely indicating a poor fit even in correctly specified models [38,39]. In this sense, the convergence of CFI, TLI, and SRMR at optimal values supports the validity of the model, although it is recommended to replicate the analysis with larger samples to confirm the stability of the fit [31,32].

### 4.3. Predictive Validity and Diagnostic Capability

The results demonstrate the scale’s predictive validity through multiple regression models, indicating its effectiveness in estimating anxiety levels and predicting the need for professional psychological help. The scale’s ability to accurately classify cases, especially those requiring psychological support, emphasizes its diagnostic utility. This helps to address the psycho-emotional issues identified in previous studies, which emphasize the need to consider their negative impact on patients with infertility issues [8,9,12,14,16].

### 4.4. Nursing Interventions and Comprehensive Care

The correlations observed between the factors on the scale are consistent with previous research, which shows that women undergoing infertility treatment have higher levels of depression, tend to use passive coping strategies, score higher on psychopathological symptoms [12], and report a lower multidimensional quality of life [16]. In this context, the SEPAR is a valuable tool, as it facilitates the identification of emotional and psychosocial needs by nurses, allowing them to formulate nursing diagnoses, set goals, and apply interventions from a comprehensive perspective. Likewise, the scale’s emphasis on perceived emotional distress, recognized as a relevant factor in increased anxiety, reinforces its usefulness in guiding specific interventions aimed at mitigating this distress and improving clinical care.

### 4.5. Implications and Future Applications

The scale’s potential impact when implemented within assisted reproduction units includes providing insights into patient needs and evaluating the effectiveness of interventions. As an essential tool for planning necessary interventions, SEPAR contributes to achieving excellence in care and empowering patients. Its comprehensive assessment aligns with the broader nursing responsibility to diagnose and intervene when functional patterns related to emotional discomfort and psychological well-being are disrupted [40].

In conclusion, the SEPAR scale offers a valuable contribution to patient-centred care in assisted reproduction units, promising enhanced understanding, targeted interventions, and, ultimately, a more holistic and humanized approach to patient care.

### 4.6. Implications for Clinical Practice

It is increasingly imperative to assess functional patterns related to the psychosocial aspects of patients, recognizing their reciprocal influence on physical and organic health. The Satisfaction and Emotional Perception in Assisted Reproduction Scale (SEPAR) offers a valuable means to measure and identify satisfaction with received care, a patient’s personal experience with assisted reproduction treatment, and their self-perception of emotional discomfort. Additionally, SEPAR enables nurses to diagnose the need for psychological consultation, empowering them to intervene in relevant functional patterns. This intervention aims to enhance both the emotional discomfort of patients and the clinical practice of nursing in the context of assisted human reproduction treatment, specifically in vitro fertilization (IVF).

Nurses can utilize this tool to identify women experiencing heightened emotional distress or diminished personal satisfaction, indicating a potential need for emotional and psychosocial intervention. In such cases, nurses can refer patients to professional psychologists if necessary. The scale’s overarching goal is to enhance communication and the patient–nurse relationship by providing information about the care received. This proactive approach aims to prevent and manage anxiety and stress associated with fertility treatment while fostering positive attitudes and realistic expectations regarding the process and outcomes.

Furthermore, the SEPAR scale assesses the potential impact of psychological or educational nursing interventions on patients’ emotional discomfort and satisfaction levels. This evaluation contributes to optimizing the care provided by nurses and underscores the scale’s utility in continuously improving patients’ experiences in assisted reproduction treatment.

It is important to note that, although the scale was initially designed for nurses, its content and structure allow it to be applied to other healthcare professionals who are directly involved in this field, such as doctors specializing in assisted reproduction. The dimensions addressed by the scale are relevant to any professional involved in these treatments. Its use by different professionals contributes to improving the quality of care and emotional support provided to these patients.

### 4.7. Strengths and Limitations

While this study contains several strengths, it is essential to acknowledge certain limitations.

Firstly, the reliability of Factor 3 (Personal satisfaction) was low (ω = 0.547) and the RMSEA was above the recommended thresholds, indicating the need to review and expand the items that comprise it to improve its internal consistency and construct validity in future applications of the scale. This factor measures personal satisfaction and consists of only two items, which may be insufficient to capture the complexity of the construct, which includes various emotional, social, and sexual dimensions. Introducing additional items could improve reliability. It would be advisable to incorporate additional items that address other facets of personal satisfaction and emotional distress, such as treatment-related stress and perceived autonomy in decision-making. The implications of this limitation are that scores may be less accurate, more sensitive to random errors, and less generalizable to other populations of patients undergoing this treatment.

Secondly, the scope of the study could be broadened by incorporating additional criteria to assess the concurrent and predictive validity of the scale. Although anxiety levels and the need for professional psychological support were relevant criteria, the absence of complementary measures limits the comprehensiveness of the scale when assessing the patient’s emotional state and satisfaction. The inclusion of other external variables would reinforce the overall validity of the scale. In this regard, the collection of external indicators related to psychological health and quality of life is an example. Increasing the sample size and collecting sociodemographic variables, such as age, socioeconomic status, educational level, or cultural context, would also contribute to broadening the applicability of the scale in clinical settings.

Thirdly, we are aware that convenience sampling has limitations; however, any research must take into account the characteristics of the context, and from a methodological point of view, this is the most appropriate approach in the healthcare context. This is justified because it facilitates the feasibility of the study in real clinical contexts, respects ethical and logistical considerations specific to healthcare, and allows useful and contextualized results to be obtained. Likewise, this type of sampling respects clinical ethics by not interfering with medical care and facilitates access to the sample, since it is more realistic to recruit patients who are present and agree to participate at the time of data collection than to propose strict random sampling. This would interrupt care or treatment processes if selected at random. Furthermore, although it is not a probabilistic sampling method, patients who agree to participate tend to share relevant characteristics with the population being treated, allowing for the collection of useful information applicable to the specific clinical context. On the other hand, convenience sampling allows for the optimization of time, personnel, and costs in an environment where access to subjects is conditioned by healthcare and administrative dynamics.

Although these limitations must be considered, they do not diminish the important contributions and practical usefulness of this study. Strengths such as originality, relevance, and the provision of a valuable tool position this research as a beneficial resource in the field of assisted reproduction treatments and psychosocial aspects.

## 5. Conclusions

The Satisfaction and Emotional Perception in Assisted Reproduction Scale (SEPAR) is a robust and dependable instrument, demonstrating its validity and reliability in measuring satisfaction with received care, personal satisfaction with assisted reproduction treatment, and the self-perception of emotional discomfort. Notably, SEPAR is beneficial due to its user-friendly nature, ease of administration, and minimal operation time required. Moreover, the scale proves effective in identifying individuals who require professional psychological support, which is a critical distinction with significant implications for patient care.

In practical terms, SEPAR equips nurses with the ability to conduct a thorough and efficient assessment of individuals undergoing fertility treatment. This streamlined approach not only aids in promptly identifying patients who need professional referrals but also contributes to the early detection of psychological disorders, ultimately enabling quicker patient recovery. Importantly, SEPAR allows nurses to pinpoint the emotional needs of patients, resulting in tailored nursing interventions aligned with key functional patterns such as sleep–rest, cognitive–perceptual, self-perception and self-concept, role–relationships, and coping–stress tolerance. If seamlessly integrated into clinical practice, SEPAR holds the potential to enhance patient care, streamline healthcare processes, and contribute to the overall well-being of those navigating assisted reproduction treatments.

## Figures and Tables

**Figure 1 healthcare-13-02416-f001:**
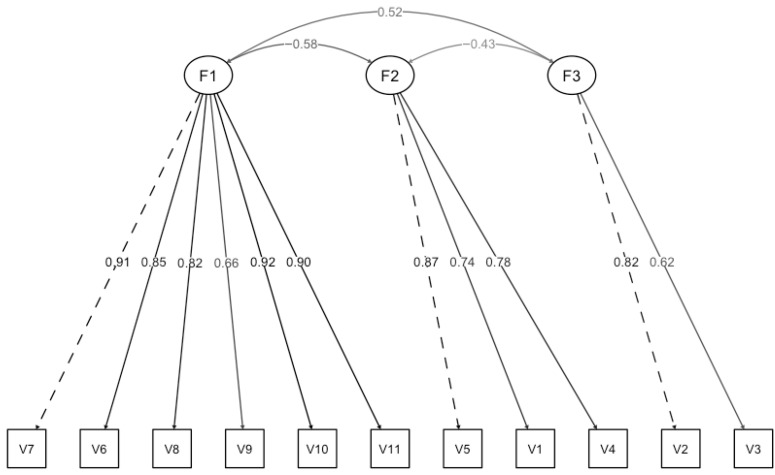
Path diagram with standardized weights.

**Figure 2 healthcare-13-02416-f002:**
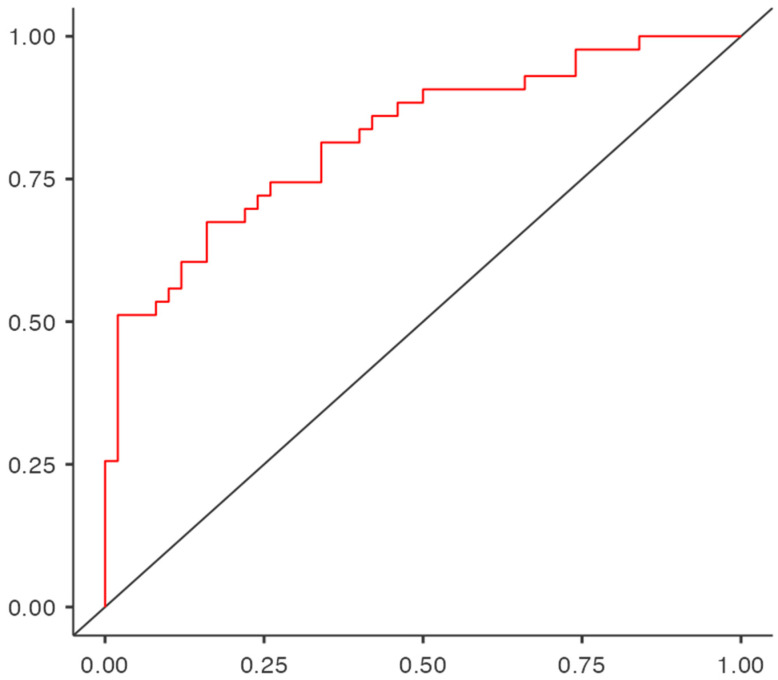
ROC curve for the Satisfaction and Emotional Perception in Assisted Reproduction Scale (SEPAR) to determine the need for professional psychological support. Note: The vertical axis represents Sensitivity, and the horizontal axis represents Specificity.

**Table 1 healthcare-13-02416-t001:** Descriptive values for each item.

Factor	Item	M	SD	DI	Sk	Ku
F1: Satisfaction with nursing care	6	5.86	2.68	0.75	−0.25	−0.54
	7	6.15	2.83	0.81	−0.60	−0.40
	8	5.23	2.86	0.80	−0.19	−0.81
	9	6.53	2.73	0.58	−0.68	−0.33
	10	4.86	2.86	0.76	0.02	−0.78
	11	5.09	2.69	0.74	−0.13	−0.51
F2: Perceived emotional discomfort	1	5.23	3.12	0.66	−0.26	−1.05
	4	2.73	3.14	0.59	0.85	−0.57
	5	5.18	3.32	0.61	−0.31	−1.26
F3: Personal satisfaction	2	7.75	2.59	0.38	−1.32	1.17
	3	8.23	1.97	0.38	−1.48	2.85

Note: Analyses performed with direct items. M = mean; SD = standard deviation; DI = discrimination index based on item–test correlation; Sk = skewness; Ku = kurtosis.

**Table 2 healthcare-13-02416-t002:** Results from multiple linear regression and logistic regression models.

	Anxiety Level				Need for Psychological Help			
	β	SE	Std (β)	CI	β	SE	Wald	OR
Satisfaction with nursing care	0.00	0.01	−0.01	[−0.18, 0.15]	−0.07	0.02	−3.00 **	0.92
Perceived emotional discomfort	8	5.23	2.86	0.80	0.10	0.03	2.84 **	1.11
Personal satisfaction	9	6.53	2.73	0.58	0.01	0.07	0.21	1.01
Constant	1.23	1.23	-	-	0.61	1.45	1.84	1.84
R^2^ adjusted	0.53				-			
R^2^N	-				0.41			

Note: ** indicate significance at the 95% level.

## Data Availability

The dataset generated and analyzed during this study can be found using the following link: https://data.mendeley.com/datasets/cx4sysywnw/1 (accessed on 9 September 2025).

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
