# Peer review of "The Development and Validation of a Satisfaction and Emotional Perception Scale for Women Undergoing Fertility Treatment"

_healthcare, 2025, doi:10.3390/healthcare13192416_

Round 1

Reviewer 1 Report

Comments and Suggestions for Authors

While the sentences about the prevalence of infertility in the first paragraph of the introduction appear to complement one another, some of them are redundant in content or repeat the same information in different sentences. I recommend rewriting this section.

The introduction to the manuscript provides a broad overview of the subject, discussing various aspects of infertility, such as its definition, prevalence, psychosocial consequences, and the importance of patient satisfaction. However, it lacks meaningful and logical transitions between paragraphs, making it difficult for readers to follow the text.

The final paragraph addresses the study's purpose. The need for this scale was not substantiated, and the study's scope was not related to the previously provided literature. I propose restructuring the introduction.

Was the study's sample size determined using power analysis? Was 97 people enough for the analysis?

Test-retest reliability is an important step in scale development studies for ensuring measurement stability (temporal reliability). Why wasn't test-retest reliability checked for this scale?

The authors stated that the initial draft of the scale had 20 items, which were later reduced to 11. However, the criteria for selecting and narrowing down the item pool were not specified.

It was stated that another scale (FertiQoL) was used as a reference during the scale development process; however, this was not adequately explained and did not have a theoretical or methodological basis.

Although the SEPAR scale's construct validity and predictive validity were supported by robust analyses, no correlation analysis was performed with a comparable scale. This is a significant limitation in terms of comparative validity. Convergent validity requires that a newly developed scale have significant relationships with an established and validated scale measuring similar constructs. For example, conducting a correlation analysis on the relationship between SEPAR and FertiQoL could have improved the scale's validity and solidified its position in the research.

Instead of providing detailed interpretations of the findings and supporting them with relevant literature, the discussion concentrated on broad statements. Statements like "the scale is valuable, practical, and useful" are frequently used, but the scientific innovation aspect of this contribution is not clearly demonstrated.

-The study found an unexpected correlation (β = 0.27) between anxiety levels and emotional well-being scores, but the discussion section does not address this contradiction.

-The RMSEA value was above acceptable levels, and the reliability of the third factor (ω = 0.55) was low, posing limitations for some dimensions of the scale. These issues should have been discussed more clearly and critically.

The study's limitations were not adequately emphasized. Issues such as sample size, factor 3 reliability, lack of test-retest, and comparative validity should have been addressed.

Author Response

We would like to express our sincere thanks to the editorial team for reviewing our manuscript and sending us their comments. The article has been modified in accordance with the helpful comments and suggestions received from the reviewers, which have helped us to improve the manuscript.

We are also very grateful for the opportunity to resubmit our work.

We have addressed all the issues raised, and the changes for reviewer no. 1, have been highlighted in red, purple for the reviewers 1 and 2, and yellow reviewers 1and 3.

Once again, many thanks.

Reviewer(s)' Comments to Author

Reviewer: 1

While the sentences about the prevalence of infertility in the first paragraph of the introduction appear to complement one another, some of them are redundant in content or repeat the same information in different sentences. I recommend rewriting this section.

We appreciate the reviewer's suggestion. We have rewritten and reorganized this section of the article. We hope it meets your expectations.

The introduction to the manuscript provides a broad overview of the subject, discussing various aspects of infertility, such as its definition, prevalence, psychosocial consequences, and the importance of patient satisfaction. However, it lacks meaningful and logical transitions between paragraphs, making it difficult for readers to follow the text.

In rewriting and reorganizing the Introduction, we have taken this suggestion from the reviewer into account. We appreciate his contribution, which has enabled us to significantly improve our work.

The final paragraph addresses the study's purpose. The need for this scale was not substantiated, and the study's scope was not related to the previously provided literature. I propose restructuring the introduction.

We thank the reviewer for their suggestion. We have taken this comment into account when rewriting and organizing the Introduction section of the paper.

Was the study's sample size determined using power analysis? Was 97 people enough for the analysis?

Sample collection was carried out using convenience sampling during the study period (9 months). This was the time available to us, given that the principal investigator was a final-year university student studying for a universtiy’s degree in nursing. This number was collected because there was not enough time to collect a higher number, provided that the criteria specified in the article were met by the participants. Clarifications regarding the reviewer's comments in the paper have been included. We thank you.

Test-retest reliability is an important step in scale development studies for ensuring measurement stability (temporal reliability). Why wasn't test-retest reliability checked for this scale?

The measurement indicated by the reviewer could not be carried out due to lack of time.

The authors stated that the initial draft of the scale had 20 items, which were later reduced to 11. However, the criteria for selecting and narrowing down the item pool were not specified.

The manuscript details the reasons that led the group of experts to remove the nine items. We appreciate your insightful comment.

It was stated that another scale (FertiQoL) was used as a reference during the scale development process; however, this was not adequately explained and did not have a theoretical or methodological basis.

We apologize to the reviewer; this paragraph is not related to this study. We have removed it.

Although the SEPAR scale's construct validity and predictive validity were supported by robust analyses, no correlation analysis was performed with a comparable scale. This is a significant limitation in terms of comparative validity. Convergent validity requires that a newly developed scale have significant relationships with an established and validated scale measuring similar constructs. For example, conducting a correlation analysis on the relationship between SEPAR and FertiQoL could have improved the scale's validity and solidified its position in the research.

The reviewer is indeed correct; however, it was decided not to implement a very lengthy protocol due to the delicate emotional and psychological situation that arises from undergoing this type of treatment.

Instead of providing detailed interpretations of the findings and supporting them with relevant literature, the discussion concentrated on broad statements. Statements like "the scale is valuable, practical, and useful" are frequently used, but the scientific innovation aspect of this contribution is not clearly demonstrated.

The discussion has been expanded and the suggestions commented on by the reviewer have been considered. We thank you for your contribution to improving our paper.

The study found an unexpected correlation (β = 0.27) between anxiety levels and emotional well-being scores, but the discussion section does not address this contradiction.

We thank the reviewer for their insightful comment. We have realised that this is not a contradiction, given that the items in factor 2 are formulated in the negative. The correlation identified between anxiety and factor 2, which indicates that the greater the emotional discomfort, the higher the level of anxiety, is logical. We have changed the name of factor 2 to make it easier to understand and, instead of emotional well-being, we have called it emotional discomfort.

The RMSEA value was above acceptable levels, and the reliability of the third factor (ω = 0.55) was low, posing limitations for some dimensions of the scale. These issues should have been discussed more clearly and critically.

The study's limitations were not adequately emphasized. Issues such as sample size, factor 3 reliability, lack of test-retest, and comparative validity should have been addressed.

We have broadened the discussion and taken your comments into account. The section on limitations has also been expanded to address the issues you have raised. We would like to reiterate our thanks to you.

Once more, we’d like to thank you for the time taken to review our paper.

The authors.

Reviewer 2 Report

Comments and Suggestions for Authors

By creating and validating the Satisfaction and Emotional Perception in Assisted Reproduction Scale (SEPAR), which can be used to gauge patient satisfaction and emotional health during fertility treatment, the manuscript makes a significant contribution to the field. Although the study's methodology is sound, there are a few areas that require improvement, especially with regard to addressing the CFA results, extending the validation criteria, and enhancing Factor 3's reliability. The scale could be a very helpful tool for doctors working in assisted reproduction with these changes, enabling more individualized and efficient patient care.

-The study uses a mixed research strategy, which works well. Nevertheless, it seems that the 97-woman sample size for the scale validation is rather small. The results may be more broadly applicable if a larger sample from several clinics is included. Clarifying the selection process used in the assisted reproduction clinics would also be beneficial, as convenience sampling may introduce biases that impact the findings. Furthermore, the scale's inclusivity would be enhanced by broadening the participant pool to encompass a wider range of demographics (such as age, socioeconomic status, and cultural background).

-It is a useful strategy that the study uses anxiety levels and the need for psychological support as criteria to test predictive validity. Further proof of the scale's usefulness, however, might come from other outside factors like psychological health evaluations or quality of life metrics. The study's findings and the scale's suitability for use in clinical settings would be strengthened by extending the criteria to evaluate concurrent validity.

-The limitations, including the limited scope of the validation criteria and the poor reliability of Factor 3, are briefly mentioned by the authors. The conversation could be extended further, though, to offer a more comprehensive examination of how these restrictions might impact the scale's practical application. Furthermore, a more thorough discussion of how these limitations affect the findings' generalizability is necessary.

Author Response

We would like to express our sincere thanks to the editorial team for reviewing our manuscript and sending us their comments. The article has been modified in accordance with the helpful comments and suggestions received from the reviewers, which have helped us to improve the manuscript.

We are also very grateful for the opportunity to resubmit our work.

We have addressed all the issues raised, and the changes for reviewer no. 2, have been highlighted in blue, purple for the reviewers 1 y 2.

Once again, many thanks.

Reviewer(s)' Comments to Author

Reviewer: 2

By creating and validating the Satisfaction and Emotional Perception in Assisted Reproduction Scale (SEPAR), which can be used to gauge patient satisfaction and emotional health during fertility treatment, the manuscript makes a significant contribution to the field. Although the study's methodology is sound, there are a few areas that require improvement, especially with regard to addressing the CFA results, extending the validation criteria, and enhancing Factor 3's reliability. The scale could be a very helpful tool for doctors working in assisted reproduction with these changes, enabling more individualized and efficient patient care.

We have considered all of the reviewer's comments and have included information in the article relating to this. We hope to meet your expectations. We appreciate your contribution to improving our work.

-The study uses a mixed research strategy, which works well. Nevertheless, it seems that the 97-woman sample size for the scale validation is rather small. The results may be more broadly applicable if a larger sample from several clinics is included. Clarifying the selection process used in the assisted reproduction clinics would also be beneficial, as convenience sampling may introduce biases that impact the findings. Furthermore, the scale's inclusivity would be enhanced by broadening the participant pool to encompass a wider range of demographics (such as age, socioeconomic status, and cultural background).

We thank the reviewer for their comments. We have included information relating to the issues they raised. We believe that their contributions have been very useful in improving our work.

-It is a useful strategy that the study uses anxiety levels and the need for psychological support as criteria to test predictive validity. Further proof of the scale's usefulness, however, might come from other outside factors like psychological health evaluations or quality of life metrics. The study's findings and the scale's suitability for use in clinical settings would be strengthened by extending the criteria to evaluate concurrent validity.

We have considered the reviewer's comment and have included text in the limitations section. We appreciate your contribution to our work.

-The limitations, including the limited scope of the validation criteria and the poor reliability of Factor 3, are briefly mentioned by the authors. The conversation could be extended further, though, to offer a more comprehensive examination of how these restrictions might impact the scale's practical application. Furthermore, a more thorough discussion of how these limitations affect the findings' generalizability is necessary.

Explanations regarding your comment have been included in the article. We appreciate the suggestions provided.

Once more, we’d like to thank you for the time taken to review our paper.

The authors.

Reviewer 3 Report

Comments and Suggestions for Authors

I appreciate the opportunity to review your study, which I find both intriguing and significant in addressing a sensitive issue within our society. Nevertheless, I would like to offer several comments that may enhance the quality of your manuscript:

  1. Comment: The abstract is too lengthy. In my opinion, it's important to indicate statistical significance and other relevant indicators, such as correlation coefficients, when presenting calculations in the results section. Additionally, the conclusion is also too long and should be summarized in 2 to 4 sentences.
  2. Comment: In lines 73-75, you discuss the challenges faced by infertile women. While the article primarily focuses on women, it would be beneficial for the introduction to acknowledge that infertile men also experience similar difficulties.
  3. Comment: Lines 108 - 112 require a citation to support these assumptions.
  4. Comment: The introduction should explain why this scale is intended for nurses. What is the role of nurses in the treatment of women with infertility?
  5. Comment: In summarizing the introduction, I would like to point out that it lacks coherence. I believe it would be beneficial to divide the introduction into several sections, such as "1.1 Infertility." This would allow us to clearly define the topics discussed in each segment, making it easier for readers to follow. The proposed sections could be organized in the following order: 1. What is infertility and what are the treatment options? 2. What psycho-emotional difficulties arise from infertility? 3. What psycho-emotional challenges are associated with infertility treatments? 4. How can patient-centered treatment and assessment of the patient's emotional state lead to better treatment outcomes? 5. What tools are currently lacking for assessing patients' psycho-emotional difficulties and their level of satisfaction? 6. What is the role of nurses in treating this patient population?
  6. Comment: What were the exclusion criteria for patients?
  7. Comment: Figures 1. The quality is quite poor; it is difficult to see the number.
  8. Comment: Same problem with the ROC curve.

I trust that my observations will provide you with valuable insights. Wishing you the best of success.

Author Response

We would like to express our sincere thanks to the editorial team for reviewing our manuscript and sending us their comments. The article has been modified in accordance with the helpful comments and suggestions received from the reviewers, which have helped us to improve the manuscript.

We are also very grateful for the opportunity to resubmit our work.

We have addressed all the issues raised, and the changes for reviewer no. 3, have been highlighted in green and yellow for the reviewers 1 y 3.

Once again, many thanks.

Reviewer(s)' Comments to Author

Reviewer: 3

I appreciate the opportunity to review your study, which I find both intriguing and significant in addressing a sensitive issue within our society. Nevertheless, I would like to offer several comments that may enhance the quality of your manuscript:

Comment: The abstract is too lengthy. In my opinion, it's important to indicate statistical significance and other relevant indicators, such as correlation coefficients, when presenting calculations in the results section. Additionally, the conclusion is also too long and should be summarized in 2 to 4 sentences.

We have rewritten the abstract taking your comments into account. We hope it meets your expectations. Thank you for your suggestion.

Comment: In lines 73-75, you discuss the challenges faced by infertile women. While the article primarily focuses on women, it would be beneficial for the introduction to acknowledge that infertile men also experience similar difficulties.

We have included a text at the end of the Introduction explaining why this work has been considered a sample of women.

Comment: Lines 108 - 112 require a citation to support these assumptions.

We have included a quote in the text you mentioned. Thank you for your comment.

Comment: The introduction should explain why this scale is intended for nurses. What is the role of nurses in the treatment of women with infertility?

We have rewritten and expanded the information in the Introduction section and explained the role of nurses in this treatment.

Comment: In summarizing the introduction, I would like to point out that it lacks coherence. I believe it would be beneficial to divide the introduction into several sections, such as "1.1 Infertility." This would allow us to clearly define the topics discussed in each segment, making it easier for readers to follow. The proposed sections could be organized in the following order: 1. What is infertility and what are the treatment options? 2. What psycho-emotional difficulties arise from infertility? 3. What psycho-emotional challenges are associated with infertility treatments? 4. How can patient-centered treatment and assessment of the patient's emotional state lead to better treatment outcomes? 5. What tools are currently lacking for assessing patients' psycho-emotional difficulties and their level of satisfaction?

  1. What is the role of nurses in treating this patient population?

The reviewer's suggestions have been taken into account, and the Introduction section has been rewritten, considering all the comments made on our work. We sincerely appreciate your help in improving our paper.

Comment: What were the exclusion criteria for patients?

The exclusion criteria have been included in the manuscript. We thank the reviewer for their insightful suggestion.

Comment: Figures 1. The quality is quite poor; it is difficult to see the number.

We have redrawn this figure and included it in the manuscript. We hope that it is now clearer and that the numbers are easier to see.

Comment: Same problem with the ROC curve.

We have redrawn the ROC curve and included it in the manuscript. We hope that it is now clearer.

Once more, we’d like to thank you for the time taken to review our paper.

The authors.

Round 2

Reviewer 1 Report

Comments and Suggestions for Authors

The authors have substantially improved the manuscript by addressing the previous revision suggestions. The scientific contribution and clarity of presentation are now at a satisfactory level. In its current form, I consider the manuscript acceptable for publication.

Reviewer 2 Report

Comments and Suggestions for Authors

Accept it